# Designing Privacy-Preserving Financial Risk Analytics on Solid Pods

Oshani Seneviratne, Fernando Spadea and Lorenzo Carta

*Rensselaer Polytechnic Institute, Troy, New York, USA*

## Abstract

We propose a decentralized, privacy-first architecture for predicting consumer financial distress, evolving beyond simulated federated environments toward a deployable, user-centric design. Leveraging Solid pods, we enforce structural data sovereignty by keeping financial data in personal storage and governing access through pod-local access control, verifiable authorization, and decentralized identity, while bringing the financial risk analytics algorithms to the data via federated learning. To mitigate inference risks from shared model updates in the federated learning process (e.g., reconstruction and membership inference), we integrate Differential Privacy via update clipping and calibrated noise injection within the user's trusted pod environment. This work demonstrates that rigorous privacy guarantees can coexist with the predictive utility required for effective, socially responsible early warning systems and intervention in consumer financial applications. Specifically, we contribute (i) a Solid-based compute-to-data architecture for financial risk modeling, (ii) a policy-governed consent lifecycle for sensitive financial data, and (iii) a layered privacy design combining structural enforcement with differential privacy.

## Keywords

Federated Learning, Financial Distress Prediction, Explainable AI, Financial Inclusion, Data Sovereignty, Responsible AI, Differential Privacy, Solid Pods, Personal Data Spaces, Consent Management, Policy Enforcement

## 1. Introduction

Financial distress is a pervasive indicator of economic instability, often culminating in adverse outcomes such as contact from debt collection agencies [1, 2]. For regulators and financial institutions, the early identification of at-risk individuals is crucial for proactive intervention and the promotion of financial inclusion. However, traditional predictive modeling relies on the centralization of sensitive personal financial data, a practice that increasingly conflicts with global privacy regulations, data sovereignty principles, and consumer trust.

In previous work [3], we addressed this challenge by developing an explainable Federated Learning (FL) framework applied to the U.S. National Financial Capability Study (NFCS) [4]. By simulating U.S. states as distinct data silos, we demonstrated that a cross-silo FL approach [5] could achieve predictive performance comparable to centralized models. Furthermore, the integration of explainable AI techniques, such as SHAP [6] and Owen values [7], allowed us to identify key distress predictors like income instability and employment gaps without inspecting raw data.

However, simulation is not deployment. While our previous model avoided centralization, it still operated on the assumption of a benevolent aggregator and static, pre-partitioned datasets. It did not account for the dynamic, user-centric nature of real-world data ownership, nor did it rigorously protect against privacy attacks, where adversaries reconstruct private data from model updates [8]. To bridge the gap between theoretical simulation and a deployable, socially responsible system, this paper proposes a novel architecture that integrates Solid for structural data sovereignty and Differential Privacy (DP) for algorithmic privacy guarantees.

In particular, the proposed design operationalizes key regulatory principles, including GDPR data minimization, purpose limitation, and privacy-by-design, by embedding consent enforcement and auditability directly into the architecture.

---

*4th Privacy & Personal Data Management Session, co-located with the 4th Solid Symposium, April 30 - May 01 2026, London, UK*

✉ senevo@rpi.edu (O. Seneviratne); spadef@rpi.edu (F. Spadea); cartal@rpi.edu (L. Carta)

🆔 0000-0001-8518-917X (O. Seneviratne); 0009-0006-4278-3666 (F. Spadea); 0009-0005-5610-9093 (L. Carta)

## 2. Solid-Enabled Data Sovereignty for Federated Financial Modeling

The core innovation of this work is the transition from simulated state-level silos to a truly decentralized, user-centric data infrastructure on Solid. Unlike traditional web applications that couple data with service logic, Solid decouples data from applications. This allows individuals to store their financial records, such as income, debt history, and demographic information, in secure, personal data stores.

Rather than the user (or an organization) uploading data to a server, the training algorithm is brought to the their data pod. The FL client operates as an authorized agent that authenticates with the user's pod. Once authenticated, the agent requests access to specific Resource Description Framework (RDF) resources governed by the user's Access Control Policies (ACP). This ensures that the model only accesses data the user has explicitly consented to share for the purpose of risk assessment. This "compute-to-data" paradigm fundamentally enhances data sovereignty.

In our financial risk modeling application, we leverage the NFCS survey data, structured as Linked Data, hosted within trusted Solid pods. The local training loop, comprising our custom model designed for categorical survey data [9], executes within each of the trusted environments.

In our prototype architecture shown in Figure 1, pods are hosted on user-controlled servers or trusted hosting providers, while FL agents execute within containerized sandbox environments with restricted network and storage access. This deployment model balances usability with isolation requirements.

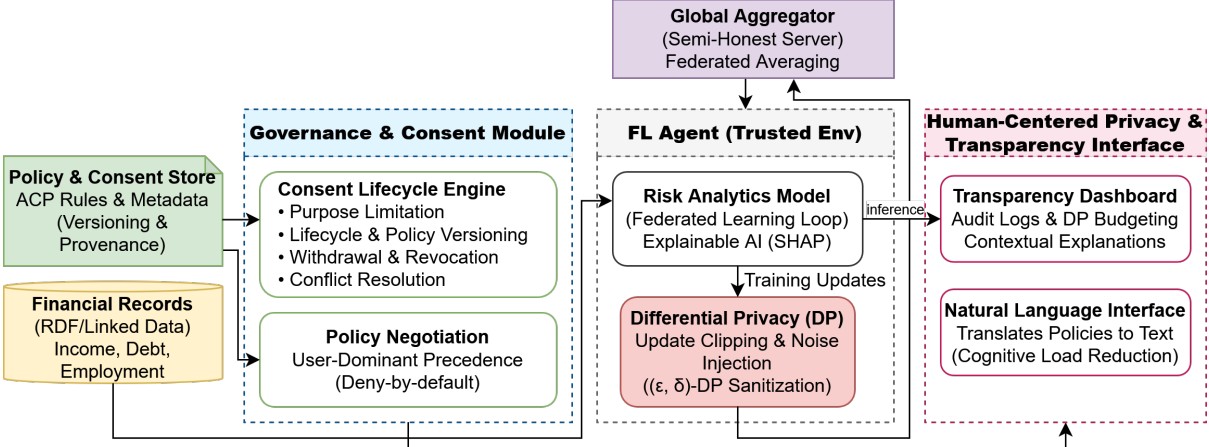

**Figure 1:** Overall architecture of the Solid-based, privacy-preserving federated financial analytics framework.

## 3. Consent and Policy Management for Solid-based Financial Data

A core requirement for privacy-preserving financial analytics is the ability to represent, enforce, and audit user consent throughout the data lifecycle. In decentralized environments such as Solid, consent must be treated as a first-class, machine-interpretable object rather than as static metadata.

**Consent Provisioning and Purpose Limitation.** In our NFCS financial distress case study, each participant grants fine-grained authorization to the FL client through Solid ACPs. Consent is expressed as a policy document that specifies (i) the authorized agent, (ii) the permitted data resources, and (iii) the intended processing purpose. The requesting agent authenticates using its WebID, and the ACP reasoner (part of the pod middleware) evaluates whether the request is compatible with the user's policy (including purpose constraints).

On the client side, the FL agent includes a declared purpose when requesting access. The pod-side middleware verifies that the declared purpose matches the policy constraint before permitting reads. This mechanism enforces purpose limitation and data minimization by construction, ensuring that model

training processes are restricted to explicitly consented use cases (e.g., denying reuse for marketing analytics or cross-context profiling).

**Consent Lifecycle and Policy Versioning.** Consent in financial contexts is dynamic: users may revise privacy preferences in response to changing risk perceptions, regulatory developments, or personal circumstances. We support explicit policy versioning to capture this evolution. Each consent artifact is assigned a persistent identifier and version number. Updates produce a new policy instance linked to prior versions using provenance metadata. Before each FL round, the agent fetches the current consent policy and validates that its cached authorization is still valid.

**Consent Withdrawal and Revocation.** Users retain the right to withdraw consent at any time. In our implementation, revocation is enacted by modifying or deleting the relevant ACP rules within the user's pod (or by publishing an explicit `revoked=true` status for the consent artifact). Once revoked, subsequent access requests from the FL client are denied.

For instance, in the NFCS case study, suppose a participant who opts out after experiencing financial recovery revokes access to their data pod. The aggregator then excludes the affected client from subsequent rounds. While previously contributed model updates cannot be retroactively removed, DP limits the long-term influence of any single participant, mitigating residual privacy risks after withdrawal.

**Policy Conflict Resolution and Negotiation.** Conflicts may arise when multiple policies apply to the same resource or when institutional requirements intersect with user-defined preferences. Our framework applies conservative precedence rules: *deny-by-default*, and *user policy dominates institutional defaults*. When conflicts occur, agents can propose less-invasive alternatives (e.g., reduced feature sets, stricter privacy budgets, or lower participation frequency). For instance, in the distress prediction scenario, a user may deny sharing raw debt-related attributes but allow participation using only coarse-grained indicators (e.g., debt-to-income buckets) or by lowering the privacy budget (smaller $\varepsilon$).

## 4. Human-Centered Consent and Transparency Interfaces

**User Comprehension of Consent and Purpose.** Financial data subjects often lack the technical expertise required to interpret ACPs, cryptographic identifiers, or privacy budgets. To address this gap, our system exposes consent decisions through layered, natural-language interfaces that translate machine-readable policies into user-facing explanations [10]. For example, rather than presenting an RDF policy document, the pod interface displays statements such as:

*"This application can access your income stability and debt indicators once per month to assess financial risk. It cannot use this data for advertising or profiling."*

These explanations are dynamically generated from policy metadata and updated when consent parameters change. This approach enables users to reason about data sharing in familiar terms while preserving formal enforcement at the protocol level.

**Permission Management and Cognitive Load Reduction.** Managing multiple applications, purposes, and privacy settings can impose substantial cognitive burden on users. In the NFCS case study, participants may simultaneously interact with budgeting tools, credit counseling services, and research-driven risk assessment agents.

To mitigate permission management overload, our design aggregates related permissions into task-oriented bundles (e.g., "financial health monitoring") and provides default, regulator-aligned templates for common use cases. Users may adopt these templates or customize them through progressive disclosure mechanisms that reveal advanced options only when needed.

Additionally, temporal controls (e.g., "allow for three months") and adaptive reminders reduce the need for continuous manual intervention, supporting sustained participation without constant policy maintenance.

**Transparency Dashboards and Feedback Mechanisms.** Meaningful consent requires visibility into how data is actually used. Our system provides pod-resident transparency dashboards that summarize recent access events, model participation history, and privacy budget. These dashboards would be generated from ACP enforcement logs and DP accounting modules, enabling users to verify compliance without inspecting raw data records or model updates. Visual indicators and anomaly alerts further support situational awareness and early detection of unexpected behavior.

**Supporting Trust and Long-Term Engagement.** By reducing complexity and increasing procedural transparency, human-centered interfaces promote trust in decentralized analytics systems. Users are encouraged to periodically review their participation status and receive contextual explanations of the societal and personal benefits derived from collective modeling. For example, participants are informed when aggregated models enable improved early intervention programs or expanded access to financial counseling services. This feedback loop reinforces perceived value and motivates continued engagement under privacy-preserving conditions.

We plan to evaluate these interfaces through usability studies and controlled experiments measuring comprehension, perceived trust, and policy management accuracy.

## 5. Structural and Algorithmic Privacy via Differential Privacy

While housing the NFCS data within Solid enforces structural privacy through pod-local access control, verifiable authorization, and decentralized identity management, FL itself is not immune to privacy attacks. Research has shown that model updates shared during the FL process can inadvertently leak sensitive information about the underlying training data including through gradient inversion, membership inference, and property inference attacks [8]. In a financial context, leaking a user's debt status or employment instability or income bracket through gradient inspection is unacceptable.

To neutralize this threat, we harden our architecture with local DP. Unlike central DP, which relies on a trusted server to add noise to the aggregated update, with local DP, the client sanitizes the update before it leaves their device in local DP. This creates a trust-minimized privacy model well-suited to the Solid ecosystem, where the aggregator is not necessarily trusted with the exact contribution of any single pod.

Our implementation integrates DP directly into the local training pipeline. After computing gradients on private data, we clip the model updates to a fixed L2 norm to bound sensitivity and limit the influence of anomalous or malicious updates and inject statistically calibrated Gaussian noise. This process constrains the information content of each contribution and reduces the risk of cumulative inference across training rounds. This mechanism ensures the output satisfies $(\epsilon, \delta)$-DP, limiting the maximum influence any single individual contribution can have on the global model while preserving aggregate utility.

Formally, DP guarantees that the distribution of observable model updates remains nearly indistinguishable with or without any single participant's data, thereby providing provable protection against reconstruction and membership inference attacks.

## 6. Threat Model and Security Considerations

Decentralized and privacy-preserving learning systems operate in adversarial environments in which participants, intermediaries, and external observers may deviate from prescribed protocols. To contex-

tualize the guarantees of our architecture, we articulate a threat model that captures relevant risks in Solid-based federated financial analytics.

**Adversarial Assumptions.** We consider a semi-honest-but-curious aggregator that follows the protocol specification but may attempt to infer sensitive information from received model updates. Participating pods are assumed to execute trusted local training code but may be exposed to compromise through malware or misconfiguration. External adversaries may observe network traffic and attempt to correlate participation patterns.

**Inference and Reconstruction Attacks.** Beyond direct access to raw data, adversaries may attempt to infer private attributes through model updates, participation patterns, or repeated query interactions. This includes gradient inversion, membership inference, and property inference attacks. Our integration of local DP mitigates these risks by bounding the information content of individual updates and limiting the distinguishability of participant contributions across training rounds.

**Malicious and Strategic Participants.** In open federated environments, some participants may behave strategically or maliciously. Relevant threats include: (i) *Model poisoning*: injecting manipulated updates to bias predictions; (ii) *Backdoor attacks*: embedding hidden behaviors in trained models; (iii) *Sybil attacks*: registering multiple pods to amplify influence; (iv) *Collusion*: coordinated behavior among subsets of participants.

In our financial case study with NFCS data, such attacks could distort risk assessments or suppress detection of vulnerable populations. While our current prototype does not fully eliminate these threats, several structural properties provide partial mitigation. Identity-bound pods, WebID-based authentication, and participation logging constrain large-scale Sybil behavior. Update clipping limits the magnitude of malicious contributions, while audit trails enable post-hoc investigation of anomalous patterns.

**Communication and Metadata Leakage.** Even when model updates are sanitized, metadata such as timing, frequency, and participation patterns may leak sensitive contextual information. For example, irregular participation may correlate with changes in financial status. Our design minimizes such leakage through batch-based training schedules and optional participation randomization, reducing the observability of individual behavior.

## 7. Conclusion

We present a blueprint for the future of privacy-preserving financial modeling on the Solid ecosystem. By evolving our previous FL simulation into a concrete architecture backed by Solid pods and DP, we demonstrate that the high stakes of financial distress prediction do not necessitate the centralization of sensitive data. Unlike conventional FL deployments, Solid provides standardized identity, policy, and interoperability primitives, enabling privacy guarantees to be embedded at the Web infrastructure level rather than implemented ad hoc within individual applications.

The integration of Solid provides the necessary structural governance, returning control to the user, while DP provides the algorithmic assurance that their data remains confidential even against sophisticated inference attacks. Together, these technologies enable a new class of *Responsible AI* applications that foster financial inclusion and regulatory compliance, proving that in the future of finance, privacy and utility can coexist.

Beyond technical privacy guarantees, our work demonstrates how Solid pods can operationalize legal, ethical, and human-centric data governance. By integrating enforceable consent, DP, and explainable analytics, we move toward an ecosystem in which individuals actively govern how their financial data contributes to collective intelligence. This positions Solid not merely as a storage infrastructure, but as a foundation for accountable, privacy-preserving socio-technical systems.

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
