# OpenReview forum: "Designing Privacy-Preserving Financial Risk Analytics on Solid Pods"
_SolidProject.org/SoSy/2026/Privacy_Session — SoSy2026-Privacy Paper_

### Official Review · ~Wout_Slabbinck1 · 2026-03-03
**Review on Designing Privacy-Preserving Financial Risk Analytics on Solid Pods**

**Rating:** 7
**Confidence:** 4

**Review:**

The short paper is relevant to the privacy session and it reads easily.
The federated financial use case and by extension differential privacy (DP) is interesting for the Solid Ecosystem. The authors also highlight several risks (malicious participants, reconstruction attacks, ...) that are still present with DP and guardrails to mitigate said risks.
For this, the authors present an architecture where a Federated Learning client acts as an agent that queries decentralized Solid pods.
End-users have then control over their data and can decide for themselves whether they want to participate and get insights or whether they want to opt-out.

However, it is unclear how the Governance and Consent module capabilities are materialized into ACP policies in the Solid pod, since many of its features cannot be handled natively by the Solid protocol.
To improve clarity, it would be helpful if the authors explain which parts can be achieved through using any Solid-compliant pod server, and which features require changes to the implementation of the server and/or sub specifications (in particular the ACP specification).
Below is a list of features that would benefit from further clarification, should the paper be accepted:
- ACP enforcement logs: In section 4 on transparancy, a dashboard makes use of ACP enforcement logs. Natively, the Solid protocol does not require servers to keep access logs and make the accessible. However, it is possible to have implementations that do keep track and allow this to be queried via an API. It would be nice if it were made clear how this is achieved.
- Temporal policy management: In section 4, the authors mention that temporal controls reduce the need for continuous manual intervention. How is this achieved with ACP? To the best of my knowledge, this is not possible natively as elaborated here [2]. To this it would be nice to elaborate how this is achieved.
- purpose constraints: In section 3 under purpose limitation, the authors state the following: "*The pod-side middleware verifies that the declared purpose matches the policy constraint before permitting reads*". Since ACP does not include constraining a permission with purposes, it would be interesting to know how this is achieved:
	- how is this encoded in the policy?
	- How is the purpose added into the request? Is this then part of the Solid-OIDC token?
- data minimization: In the same paragraph, data minimization is mentioned. This is also not part of Solid protocol. Neither in the ACP specification nor in the description of the LDP Resources interface aspect as well.
	- Which strategy is employed to indicate that a subselection of data can be accessed? Is it similar to Derived Rersouces as proposed by Van Herwegen & Verborgh [3]?
- consent: The authors claim that their design embeds consent enforcement. In section 3, they elaborate that this is done through a policy document and it is hinted that is achieved through ACPs. However, Florea & Esteves showed in [1] that currently automated consent under GDPR in Solid can not be achieved. If the authors use something else than ACP, it would be advised to declare explicit how consent is achieved.
- Policy conflict resolution
	- ACP does indeed works with a deny-by-default strategy, furthermore it employs deny over allow. However,  without further details it is unknown how "*user policy dominates institutional defaults*" is achieved using ACP.



[1] Florea, M., & Esteves, B. (2023). Is Automated Consent in Solid GDPR-Compliant? An Approach for Obtaining Valid Consent with the Solid Protocol. _Inf._, 14, 631. [https://doi.org/10.20944/preprints202307.1344.v2](https://doi.org/10.20944/preprints202307.1344.v2).
[2]W. Slabbinck, J. A. Rojas, B. Esteves, R. Verborgh, and P. Colpaert, ‘Enforcing Usage Control Policies in Solid using Rule-Based Web Agents’, in _Proceedings of the Posters and Privacy Session of the Solid Symposium 2024_, B. Esteves, J. Hofmann, and S. Schmid, Eds, in CEUR Workshop Proceedings, vol. 3947. Leuven, Belgium: CEUR, May 2024, pp. 109–117. Accessed: Apr. 07, 2025. [Online]. Available: [https://ceur-ws.org/Vol-3947/short15](https://ceur-ws.org/Vol-3947/short15)
[3] J. Van Herwegen and R. Verborgh, “Granular Access to Policy-Governed Linked Data via Partial Server-Side Query,” in _Proceedings of the 21st Extended Semantic Web Conference: Posters and Demos_, 2024, pp. 331–335.

---

### Official Review · ~Víctor_Rodríguez-Doncel1 · 2026-03-05
**No demo, but still, a great idea well described**

**Rating:** 9
**Confidence:** 3

**Review:**

Some might thing that "federated learning" is an old-fashioned topic, but I think the paper presents a great idea: the marriage of Solid Pods with differential privacy - Solid enforces who controls the data, and differential Privacy protects how the data is used.

The proposed architecture maximizes data sovereignty (true user control), eliminates unnecessary data transfers, and the enforceable, machine-readable consent solves the trust problem better than centralized AI ever can. The paper is well writen and balanced. Also, the Threat Model and Security Considerations is not a systematic analysis (like using NIST SP 800‑154 would be), but still more than enough for this purpose.

The only thing I miss is an implementation or demo --there is no github repo.

Final comment from my "EU-perspective": this schema is also relevant for implementing systems compliant with a number of EU regulations: GDPR (and the benefit is threefold: data minimization, purpose limitation, portability), AI Act (better risk-based safeguards), Data Governance Act (the FL is a data aintermediary, data altruism is possible without compromising personal data) or the Digital Markets Act.

(I tried not to be impressed by the fact that the first author leads a great research group, and herself was a phd student of Tim Berners-Lee: the paper was great to me!)

---

### Decision · Program_Chairs · 2026-03-09

Accept (Paper)